# Baseline Sensitivity and Control Efficacy of Various Group of Fungicides against *Sclerotinia sclerotiorum* in Oilseed Rape Cultivation

Nazanin Zamani-Noor 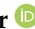

Julius Kühn-Institute, Institute for Plant Protection in Field Crops and Grassland, Messeweg 11-12, D-38104 Braunschweig, Germany; nazanin.zamani-noor@julius-kuehn.de; Tel.: +49-531-2994530

**Abstract:** Sclerotinia stem rot (SSR), caused by *Sclerotinia sclerotiorum*, is a devastating disease of oilseed rape that may cause significant yield losses if not controlled by cultural management strategies and fungicide applications. Studies were conducted to evaluate the efficacy of different group of fungicides as well as a biopesticide, including azoxystrobin, boscalid, fludioxonil, prothioconazole, tebuconazole, azoxystrobin/tebuconazole, boscalid/pyraclostrobin, prothioconazole/fluopyram and *Bacillus amyloliquefaciens*, on baseline sensitivity of *S. sclerotiorum* isolates under in-vitro conditions as well as control of SSR in the field. Artificial inoculation and mist irrigation prompt to reproducible SSR infection in oilseed rape cultivation. All compounds significantly reduced 36.7% to 86.9% SSR severity and increased 55.2% to 98.7% yield, 1.5% to 7.0% thousand grain-weight, 1.5% to 5.9% oil and 0.1% to 1.3% protein content. Fludioxonil, boscalid/pyraclostrobin and fluopyram/prothioconazole achieved strongest fungicidal activity against SSR. The biopesticide provided 36% of disease control. Under in vitro conditions, *B. amyloliquefaciens* not only strongly inhibited mycelial growth but also the formation of sclerotia in all concentrations. Boscalid and fludioxonil exhibited the highest level of fungicidal activity against *S. sclerotiorum*, with mean EC50 values of 1.23 and 1.60 µg a.s. mL$^{-1}$. The highest variability of EC50 values between isolates was observed towards prothioconazole and azoxystrobin.

**Keywords:** sclerotinia stem rot; *Brassica napus*; *Bacillus amyloliquefaciens* strain QST 713; fungicide efficacy; biopesticide; biological control; mycelial growth inhibition; EC50

## 1. Introduction

Sclerotinia stem rot (SSR), caused by *Sclerotinia sclerotiorum* (Lib.) de Bary, is a disease of major importance worldwide. The pathogen has a remarkably broad host rang, which can infect more than 400 plant species, including economy important arable crops such as oilseed rape, mustards, soybean, and sunflower [1]. The fungus leads to stem rot of oilseed rape (*Brassica napus* L.), which is a very damaging disease that occurs in all oilseed rape-growing areas in Germany. The pathogen can infect oilseed rape crops in two different ways, either basal infection in autumn from myceliogenically germinated sclerotia (happens seldom) or in spring through ascospores that come from carpogenic germination of sclerotia (main way of infection). Flower petals of oilseed rape supply as an energy source to help ascospore infection of healthy plants [2]. Once disease is established, it can expand to stem, leaves, and pods. Central to the success of *S. sclerotiorum* as a plant pathogen is the production of several sclerotia on infested plants. In addition to characterizing an extensive reproductive capability and providing the primary inoculum for following epidemics, these sclerotia facilitate the pathogen to survive over winter. Previous study demonstrated that not only the number of sclerotia inside the soil is an essential factor in disease outbreak but also the sclerotial size and weight influenced the germination rate of sclerotia and production of ascospores [3]. It was observed that oilseed rape plants potentially return more sclerotia by number or weight to the soil in comparison with the other crops tested

in their study and hence potentially pose a bigger disease risk for following susceptible plants [3].

To date, no Sclerotinia-resistant oilseed rape cultivar is available in German market. Therefore, integrated control of SSR relies on the cultural management strategies and fungicide applications. One time application of fungicides in spring during plant flowering stage (BBCH 60–69; BBCH monograph, [4]) appears to provide the best effective control of the disease. Various group of fungicides such as pyridine- carboxamide (succinate- dehydrogenase inhibitors; SDHI), triazole (de-methylation inhibitors; DMI), and strobilurin (quinone outside inhibitors; QoI) have been extensively used to control SSR disease over the past decade in Germany, but widespread control failures were reported in some years. Nowadays all mentioned groups of fungicide show a risk of developing resistance and were listed as a medium to high-risk fungicides for evolution of resistance by the Fungicide Resistance Action Committee [5]. Resistance to SDHI fungicide boscalid has been shown in field isolates of *Alternaria alternata* [6] and *Botrytis cinerea* [7]. Additional studies suspected that boscalid sensitivity have decreased in field isolates of *S. sclerotiorum* [8–10]. In contrast, Hu et al. [11] studied baseline sensitivity of *S. sclerotiorum* to boscalid by determining the effective concentration causing mycelial growth inhibition by 50% (EC50) values of different isolates in different years. They observed no boscalid sensitivity shift in *S. sclerotiorum* isolates between 2008 and 2014. Further Sclerotinia sensitivity evaluation to DMIs (e.g., tebuconazole and prothioconazole) and QoIs (e.g., azoxystrobin and pyraclostrobin) demonstrated that all studied isolates were sensitive to the tested fungicides and no resistant isolates have been detected in the fungal populations in different countries [12–15].

In parallel, using microbial agents to control SSR can be an eco-friendly and cost-effective substance of an integrated management program. In the search for alternatives to fungicides, several microorganisms have shown effective as potential biocontrol agents for controlling the disease. To date, the most promising biocontrol agent, *Coniothyrium minitans* [16], has been widely used for destroying sclerotia in the soil. As part of the mycoparasitic process, *C. minitans* produces cell-wall degrading enzymes such as chitinases and glucanases, which are responsible for cell wall degradation and host tissue collapse in *S. sclerotiorum* [17]. Furthermore, several endophytes such as *Bacillus subtilis* and other closely related *Bacillus* spp. have taken in consideration for development of biological control agents. These species are able to produce various kinds of antibiotics, which inhibit the mycelial growth and spore germination of fungal pathogens [18].

Nevertheless, although a high level of resistance to fungicides has not yet been shown in field isolates of *S. sclerotiorum*, it is far crucial to achieve competent fungicides with various modes of action to replace the compounds that are presently applied to control SSR. In addition, it is necessary to evaluate combination of alternative modes of action to reduce selection for fungicide resistance to elevate disease management.

The specific objectives of the present study are to: (i) Evaluate the efficacy of a range of fungicides in different classes as well as a biopesticide against Sclerotinia stem rot in oilseed rape cultivation under field conditions; (ii) determine the effect of foliar applications of fungicides on yield and yield parameters of oilseed rape; (iii) in vitro screening sensitivity of *S. sclerotiorum* isolates to different concentrations of several fungicides with various mode of actions; and (iv) assess effects of different groups of fungicides on sclerotial formation.

## 2. Materials and Methods

### 2.1. Efficacy Evaluation of Fungicides for Control of Sclerotinia Stem Rot under Field Conditions

2.1.1. Experimental Method

Field trials were conducted over two consecutive years in Braunschweig (Bundesallee), Lower Saxony, Germany in 2018/2019 and 2019/2020. The winter oilseed rape cv. Avatar (Norddeutsche Pflanzenzucht Hans-Georg Lembke KG, Holtsee, Germany) was chosen due to its susceptibility to SSR (intern communication with breeder). Seeds were drilled in 22.5 cm rows on 28 August 2018 and 5 September 2019 respectively, at a seeding rate of

60 seeds per m$^2$. The experiments were conducted in a complete randomized block design with four replications. The plot size was $10 \times 4.5$ m.

Good weeds and insect control in all experiments were achieved by applying appropriate herbicides (e.g., Butisan Gold and Runway at BBCH 10–18) and insecticides (e.g., Trebon 30 EC at BBCH 14–16 and Biscaya at BBCH 60–69). To increase the incidence and severity of SSR, field's plots were artificially inoculated.

### 2.1.2. Inoculums Preparation and Plant Inoculation

Different isolates of *S. sclerotiorum*, which were collected as sclerotia from naturally Sclerotinia-infected oilseed rape crops in different geographic regions in Germany between 2017 and 2019, were used to prepare the inoculums. Stocks of each isolate were preserved as sclerotia kept at 4 °C and new cultures were triggered by placing a sclerotium on potato dextrose agar (PDA; Carl Roth GmbH, Karlsruhe, Germany) and incubating at room temperature for five days to produce actively growing cultures for preparing inoculums. The wheat seed inoculum was prepared by placing 2 kg of wheat grains into autoclavable plastic bags (60 L, Carl Roth GmbH, Karlsruhe, Germany) and then hydrated with distilled water for 24 h, the extra water was poured off, and the grains were autoclaved twice. Later on, several sclerotinia mycelial plugs from the margin of colonies of each isolate were transferred into each bag. Inoculated bags were then incubated at room temperature for two weeks during which the bags were shaken 2–3 times weekly to ensure equal distribution of the inoculum. Infested wheat seeds were then air-dried for 3 days under room temperature and ground roughly with cereal laboratory mill. The inoculum was preserved at 4 °C until required. When the plants reached the growth stage 64–65 (40–50% flowers on main raceme open, older petals falling; BBCH monograph, [4]), 50 g dry inoculums per m$^2$ was spread evenly on the canopy of the oilseed rape plants by hand. In each experimental field, non-inoculated and non-treated control plots were left in order to check the presence and consistency of natural inoculum. Directly after inoculation, 10 mm irrigation water was applied equally to all plots every other day until 10 days post inoculation (dpi).

### 2.1.3. Fungicide Application

One day after inoculation (BBCH 64–65), all fungicides as well as the biopesticide, *Bacillus amyloliquefaciens* strain QST 713, were applied at recommended dose rate to the plant canopy using a tractor-mounted plot-sprayer at 300 L/ha. The complete treatment schedule and detailed information about fungicides used in this study are summarized in Table 1.

**Table 1.** Fungicides evaluated under field and in vitro conditions against *Sclerotinia sclerotiorum* the causal agent of Sclerotinia stem rot in oilseed rape cultivation.

| Commercial Product | Active Substance (a.s.) | a.s. Content | Application Rate | Mode of Action | Manufacture | Field (F)/In Vitro (I) |
|---|---|---|---|---|---|---|
| Ortiva | Azoxystrobin | 250 g/L | 0.5 kg/ha | QoI | Syngenta Agro GmbH | F and I |
| Cantus | Boscalid | 500 g/kg | 1.0 L/ha | SDHI | BASF SE | F and I |
| Treso | Fludioxonil | 500 g/kg | 0.75 kg/ha | Inhibitor of MAP | Syngenta Agro GmbH | F and I |
| Proline | Prothioconazole | 250 g/L | 0.7 L/ha | DMI | Bayer Crop Science | I |
| Folicur | Tebuconazole | 250 g/L | 1.5 L/ha | DMI | Bayer Crop Science | F and I |
| Custodia | Azoxystrobin + Tebuconazole | 120 gL 200 g/L | 1.0 L/ha | DMI + QoI | ADAMA Deutschland GmbH | F |
| Pictor Active | Boscalid + Pyraclostrobin | 150 g/L 250 g/L | 1.0 L/ha | SDHI + QoI | BASF SE | F |

**Table 1.** *Cont.*

| Commercial Product | Active Substance (a.s.) | a.s. Content | Application Rate | Mode of Action | Manufacture | Field (F)/In Vitro (I) |
|---|---|---|---|---|---|---|
| Propulse | Fluopyram + Prothioconazole | 125 g/L 125 g/L | 1.0 L/ha | DMI + SDHI | Bayer Crop Science | F |
| Serenade ASO | *Bacillus amyloliquefaciens* * strain QST 713 | 13.9 g/L ($10^{12}$ cfu/kg) | 2.0 L/ha | Microbial | Bayer Crop Science | F and I |

* Former name: Bacillus subtilis.

### 2.1.4. Sclerotinia Disease Assessment

The field trials were assessed visually for Sclerotinia stem rot between mid to late June approximately at the BBCH growth stage 80–83, when 10–30% of pods ripped and seeds were dark and hard [4]. One hundred plants were collected randomly in each plot rated for SSR on a 0–3 scale, where 0 = no symptoms, 1 = up to 25% of the stem circumference girdled by lesions; 2 = between 25% and 50% of the stem circumference girdled by the lesions; 3 = plants almost dead. Disease severity index (DSI) was calculated from each infection types according to the following formula:

$$\text{Disease severity index} = \frac{\sum(\text{n0} \times 0 + \text{n1} \times 1 + \text{n2} \times 2 + \text{n3} \times 3)}{\text{N} \times \text{No.Classes with symptoms}} \times 100. \quad (1)$$

In which, n is number of plants in each class, N is the total number of plants and 0, 1, 2 and 3 are the symptom severity classes.

Efficacy (%) of each fungicide treatment for control of SSR was calculated using following formulate:

$$\text{Efficacy (\%)} = 100 - \left(\frac{\text{T}}{\text{C}} \times 100\right). \quad (2)$$

where T is the value of DSI of fungicide treatment and C is value of DSI of non-treated inoculated control. The mean of the percentage of the four variables was used to provide the percentage of overall efficacy.

### 2.1.5. Yield and Seed Quality

Plots were harvested on 22 July 2019 and 15 July 2020, respectively, with a Haldrup-C58 plot combine-harvester (Haldrup GmbH, Ilshofen, Germany) and the seed yield was determined at water content of 9% [19]. Subsequently, a homogenous 500 g sample of seeds was taken from each plot, and the weight (g) of thousand grains (TGW) was recorded. The relative yield (%) was further calculated as following: (yield of treated plot/yield of untreated control) × 100.

Additionally, near-infrared reflectance spectroscopy (NIRS) was used to develop calibration models for the prediction of protein and oil content in seeds. Protein and oil contents were assessed in percent at 91% seed dry matter [20].

### 2.1.6. Statistical Analysis

The disease severity of Sclerotinia stem rot of the non-inoculated and untreated control was less than 2% in both trials; therefore, these data were not included in the analyses. In the current study, the plots that were artificially inoculated with *S. sclerotiorum* without fungicide treatments were used as control to evaluate the fungicide efficacy. Comparisons between different treatments and disease severity index were performed by analysis of variance (ANOVA) using Fisher's least significant difference (LSD) and considered significant at $p \leq 0.05$ in Statistica version 9.1 (Stat Soft, Inc., Tulsa, OK, USA). Because variances were homogeneous ($p \leq 0.05$), combined data sets were then analyzed. Years, replications, and all possible interactions containing these effects were considered as

random effects and fungicides were considered as fixed effects. Considering year as an environmental or random effect permits inferences about treatments to be made over a range of environments [21]. Spearman's rank correlation coefficient was used to analyze the relationship between Sclerotinia disease severity with yield, TGW, as well as oil and protein content.

### 2.2. In Vitro Sensitivity Evaluation of S. sclerotiorum Isolates to Fungicides

### 2.2.1. Fungicide Sensitivity Screening

Fungicide sensitivity of *S. sclerotiorum* isolates to different group of fungicides (Table 1) was carried out in vitro by analyzing the distribution of EC50 values (50% effective concentration) and percent of growth inhibition of 68 isolates. The isolates were collected from naturally Sclerotinia-infected oilseed rape fields in different geographical regions in Germany during 2017 to 2019. Sensitivity tests were conducted using fungicide-amended potato dextrose agar (PDA) plates at 0.0, 0.1, 0.3, 1.0, 3.0, 10.0, 30.0, 100.0 µg a.s. $mL^{-1}$ concentrations. Mycelial plugs were cut with a 5-mm diameter cork-borer (Carl Friedrich Usbeck KG, Radevormwald, Germany) from the margin of three days colonies of each isolate and placed with mycelium-side down on the centers of fungicide-amended PDA. Plates were then incubated in the dark at 20 °C. Each treatment was represented by three replicate Petri dishes, and the experiment was repeated twice. Colony diameters (minus the diameter of the inoculation plug) were measured in two directions at 90° to one another at 3 days post inoculation (dpi). The mean growth values were obtained and then converted into the mycelial growth inhibition (%) in relation to the control treatment by using the Abbott's formula:

$$\text{Mycelial growth inhibition (\%)} = \frac{C - T}{C} \times 100$$

where, *C* and *T* represent mycelial growth diameter (mm) in control and treated Petri dishes, respectively.

Effective fungicide concentration to inhibit mycelial growth by 50% (EC 50, µg/mL) were further calculated according to linear regression of colony diameter on log-transformed fungicide concentration using the mean colony diameter for all replicates at each concentration [22].

### 2.2.2. Effect of Fungicides on Sclerotia Formation

Inoculated plates from previous study were further incubated in 25 °C in darkness for 21 days for sclerotia formation. The number of sclerotia formed on each Petri dish were counted manually. Furthermore, all sclerotia in each plate were weighed after being dried for additional three days at 25 °C in darkness.

### 2.2.3. Statistical Analysis

Comparisons between fungicide sensitivities were made by ANOVA. Treatments (isolates and fungicides) and experiment effects were considered as random factors in the analysis. Means were compared with Fisher's protected least significant difference test at $p \leq 0.05$ in Statistica version 9.1 (Stat Soft, Inc., Tulsa, OK, USA).

## 3. Results

### 3.1. Fungicide Efficacy Evaluation under Field Conditions

Sclerotinia stem rot severity of the non-inoculated, untreated controls were less than 2% in both trials in both years; therefore, these data were not included in the analyses. The artificial inoculation and mist irrigation resulted in severe SSR in control plots up to 80% disease severity index (Table 2).

**Table 2.** Effect of various group of fungicides as well as a biopesticide in controlling Sclerotinia stem rot in oilseed rape cultivation under field conditions.

| Treatments | DSI ± SD [1,2] (%) | Efficacy of Fungicide [3] (%) | Yield ± SD [1] (dt/ha) | Yield Relative to UTC [4] (%) | TGW ± SD [1] (g) | TGW Relative to UTC [4] (%) | Oil Content ± SD [1] (%) | Oil Relative to UTC [4] (%) | Protein Content ± SD [1] (%) | Protein Relative to UTC [4] (%) |
|---|---|---|---|---|---|---|---|---|---|---|
| Inoculated-untreated control | 82.5 ± 4.8 [a] | - | 19.6 ± 4.7 [a] | 100 | 5.2 ± 0.2 [a] | 100 | 40.8 ± 1.3 [a] | 100 | 20.0 ± 1.5 [a] | 100 |
| Azoxystrobin | 35.3 ± 17.1 [c] | 57.2 | 34.6 ± 2.1 [c] | 176.4 | 5.5 ± 0.1 [bc] | 105.3 | 42.9 ± 0.8 [b] | 105.3 | 20.1 ± 0.5 [a] | 100.2 |
| Boscalid | 32.2 ± 5.6 [c] | 60.9 | 35.8 ± 2.2 [cd] | 182.3 | 5.6 ± 0.1 [c] | 107.0 | 42.8 ± 0.5 [b] | 104.9 | 20.3 ± 0.7 [ab] | 100.7 |
| Fludioxonil | 10.8 ± 7.1 [e] | 86.9 | 37.6 ± 2.4 [cd] | 191.5 | 5.4 ± 0.1 [abc] | 104.9 | 43.2 ± 0.9 [bc] | 105.9 | 20.1 ± 0.8 [a] | 100.1 |
| Tebuconazole | 34.4 ± 7.2 [c] | 58.3 | 33.8 ± 3.9 [bc] | 172.0 | 5.3 ± 0.3 [ab] | 101.5 | 41.7 ± 1.8 [a] | 102.3 | 20.4 ± 0.9 [b] | 100.2 |
| Azoxystrobin + Tebuconazole | 25.3 ± 7.3 [cd] | 69.7 | 38.9 ± 1.2 [d] | 197.9 | 5.5 ± 0.2 [bc] | 105.3 | 42.0 ± 0.9 [b] | 103.1 | 20.2 ± 1.1 [a] | 100.4 |
| Boscalid + Pyraclostrobin | 12.8 ± 5.2 [e] | 84.5 | 39.0 ± 1.1 [d] | 198.7 | 5.5 ± 0.2 [bc] | 105.3 | 42.7 ± 1.5 [b] | 104.7 | 20.5 ± 0.9 [b] | 101.3 |
| Fluopyram + Prothioconazole | 17.5 ± 6.8 [d] | 78.8 | 37.1 ± 1.6 [cd] | 189.0 | 5.5 ± 0.1 [bc] | 105.3 | 43.3 ± 0.7 [bc] | 105.5 | 20.1 ± 0.6 [a] | 100.0 |
| *B. amyloliquefaciens* strain QST 713 | 52.2 ± 7.1 [b] | 36.7 | 30.5 ± 3.0 [b] | 155.2 | 5.5 ± 0.1 [bc] | 105.3 | 41.4 ± 0.6 [a] | 101.5 | 20.1 ± 0.7 [a] | 100.1 |

[1.] Each value is the mean of two independent experiments in 2018/2019 and 2019/2020 with four replicates ± standard deviations. Means in a column followed by the same letter were not different according to Fisher's least significant difference (LSD) ($p \leq 0.05$); [2.] the Sclerotinia infection type on each plant was visually determined based on a 0–3 scale; disease severity index (DSI) was calculated from each infection type. [3.] Efficacy of fungicide (%) = 100—[(DSI of treated plot/DSI of untreated control) × 100]. [4.] Relative values = (mean value of treated plot/mean value of untreated control) × 100; artificially inoculated but untreated controls were set to 100.

All fungicide treatments as well as biopesticide, *B. amyloliquefaciens* strain QST 713, reduced SSR and yield loss in comparison with inoculated but untreated controls, and there were significant differences among the treatments (Table 2). The mean overall efficacy was also different among fungicides. Fludioxonil and combinations of boscalid/pyraclostrobin and fluopyram/prothioconazole were the most effective fungicides, significantly reducing Sclerotinia disease severity down to 10.8%, 12.8% and 17.5%, respectively (Table 2). Disease severity indices of plots treated with azoxystrobin/tebuconazole, boscalid, tebuconazole and azoxystrobin ranged between 25% and 35% (Table 2). Furthermore, the SSR was decreased to 52% by application of the biopesticide; *B. amyloliquefaciens* strain QST 713 (Table 2).

Fungicide treatments exhibited significant differences in yield (dt/ha) and TGW (g). The lowest values of seed yield (19.6 dt/ha) and TGW (5.2 g) were recorded from inoculated but untreated plots and was significantly lower than all plots with pesticide treatments (Table 2). Among all compounds tested, application of *B. amyloliquefaciens* strain QST 713 produced the lowest seed yield by 30.5 dt/ha but TGW was not significantly different with other fungicides (Table 2). Highest value of yield (dt/ha) was achieved by the application of boscalid/pyraclostrobin mixture with 39.0 dt/ha followed by application of azoxystrobin/tebuconazole with 38.9 dt/ha and fludioxonil with 37.6 dt/ha (Table 2). The highest TGW (5.6 g) were achieved by the application of boscalid, the lowest value (5.3 g) was recorded in plots treated with tebuconazole (Table 2).

Significant differences among fungicide treatments were detected for oil and protein contents (Table 2). Treatments that did not provide a significant oil increase over the inoculated untreated controls were tebuconazole and *B. amyloliquefaciens* strain QST 713 (Table 2). Fungicide treatments that significantly increased protein content compared to inoculated untreated controls were tebuconazole and the combination of boscalid/pyraclostrobin (Table 2).

The results of Spearman's rank correlation coefficient revealed a significant, negative relationship between SSR and yield, TGW and oil content. A stronger relationship was observed between disease severity index and total yield ($r_s = -0.96$, $p < 0.001$) than between DSI and TGW ($r_s = -0.63$, $p = 0.045$) and between DSI and oil content ($r_s = -0.86$, $p = 0.003$). In contrast, the protein content was not significantly correlated with SSR severity at $p \leq 0.05$ ($r_s = -0.33$, $p = 0.382$) (Table 3).

**Table 3.** Spearman's rank correlation coefficient between Sclerotinia stem rot severity and yield as well as thousand grain weight (TGW), oil and protein content.

| Variables | Linear Equation | $r_s$ | *p*-Value |
|---|---|---|---|
| Sclerotinia disease severity/Yield | y = 42.8734 − 0.2603x | −0.96 | <0.001 * |
| Sclerotinia disease severity/TGW | y = 5.5531 − 0.0033x | −0.63 | 0.045 * |
| Sclerotinia disease severity/Oil content | y = 43.4102 − 0.0322x | −0.86 | 0.003 * |
| Sclerotinia disease severity/Protein content | y = 20.3088 − 0.0029x | −0.33 | 0.382 [ns] |

* correlation is significant at $p \leq 0.05$; ns = not significant.

### 3.2. In-Vitro Sensitivity Evaluation of S. sclerotiorum Isolates

3.2.1. Inhibition of Hyphal Growth

Analysis of variance showed differences in the inhibition of mycelial growth among fungicides and the effect of them, *S. sclerotiorum* isolates and their interaction were all significant ($p \leq 0.05$). PDA amended with each of the five fungicides as well as biopesticide *B. amyloliquefaciens* strain QST 713 reduced mycelial growth of *S. sclerotiorum* in comparison with the growth on non-amended medium (Table 4). Under in-vitro conditions, *B. amyloliquefaciens* strain QST 713 was the most effective compound against *S. sclerotiorum* isolates because no mycelia growth were observed over all concentrations (Table 4). At a concentration of 0.1 μg a.s. mL$^{-1}$, mycelial growth inhibition of *S. sclerotiorum* among tested fungicides ranged from 6.6% to 25% (Table 4). Growth of the pathogen was reduced by 51% for fludioxonil at 0.3 μg a.s. mL$^{-1}$, whereas growth suppression in the presence

of azoxystrobin, boscalid, prothioconazole, and tebuconazole at the same rate was significantly less (12% to 27%). At 1.0 and 3.0 μg a.s. mL$^{-1}$, boscalid and fludioxonil suppressed mycelial growth of the *S. sclerotiorum* isolates from 47% to 65% and 66% to 79%, whereas the activity of azoxystrobin, prothioconazole and tebuconazole ranged from 20% to 42% and 43% to 61% respectively. At a concentration of 10 μg a.s. mL$^{-1}$, fludioxonil was followed by prothioconazole and boscalid, which restricted *S. sclerotiorum* growth from 78% to 90%, whereas reduction of growth by azoxystrobin and tebuconazole was significantly less from 55% to 72% (Table 4). At rate of 30 μg a.s. mL$^{-1}$, inhibition of mycelial growth of *S. sclerotiorum* ranged from 88% to 98% for all tested fungicides except azoxystrobin (Table 4). At a rate of 100.0 μg a.s. mL$^{-1}$, inhibition of growth of this fungus ranged from 91% to 99% for fludioxonil, prothioconazole, tebuconazole, and boscalid, whereas reduction of growth by azoxystrobin was significantly less at 66% (Table 4).

**Table 4.** Mycelial growth inhibition ± SD of *Sclerotinia sclerotiorum* isolates (*n* = 68) grown on potato dextrose agar (PDA) amended with five fungicides as well as a biopesticide at seven concentrations [a].

| Fungicides | Concentration (μg a.s. mL$^{-1}$) | | | | | | |
| --- | --- | --- | --- | --- | --- | --- | --- |
| | 0.1 | 0.3 | 1 | 3 | 10 | 30 | 100 |
| Azoxystrobin | 9.0 ± 8.3 [a] | 26.9 ± 18.5 [a] | 38.0 ± 23.9 [b] | 50.4 ± 22.7 [a] | 55.8 ± 25.1 [a] | 61.2 ± 24.2 [a] | 66.3 ± 19.2 [a] |
| Boscalid | 6.6 ± 7.7 [a] | 23.0 ± 13.1 [a] | 46.7 ± 10.0 [bc] | 66.4 ± 7.2 [b] | 78.8 ± 7.7 [ab] | 88.7 ± 5.9 [b] | 91.7 ± 6.3 [b] |
| Fludioxonil | 25.9 ± 24.8 [b] | 50.9 ± 35.5 [b] | 65.1 ± 27.1 [c] | 79.0 ± 18.7 [bc] | 91.9 ± 8.3 [b] | 98.4 ± 1.7 [b] | 99.8 ± 0.5 [b] |
| Prothioconazole | 14.7 ± 24.2 [ab] | 19.9 ± 24.4 [a] | 42.9 ± 29.0 [b] | 61.9 ± 18.5 [ab] | 84.5 ± 12.1 [b] | 91.1 ± 7.3 [b] | 98.1 ± 3.8 [b] |
| Tebuconazole | 7.8 ± 17.9 [a] | 12.3 ± 24.1 [a] | 19.9 ± 31.1 [a] | 43.3 ± 35.0 [a] | 72.1 ± 28.2 [a] | 88.9 ± 16.5 [b] | 97.3 ± 4.4 [b] |
| *B. amyloliquefaciens* | 100 ± 0.0 [c] | 100 ± 0.0 [c] | 100 ± 0.0 [d] | 100 ± 0.0 [c] | 100 ± 0.0 [b] | 100 ± 0.0 [b] | 100 ± 0.0 [b] |

[a] Mycelial growth inhibition (%) = [(the average diameter of the fungal colony in the control—the average diameter of the treated colony)/the average diameter of the fungal colony in the control] × 100; means in a column followed by the same letter were not different according to Fisher's least significant difference (LSD) ($p \leq 0.05$).

EC50 values for boscalid and fludioxonil were the lowest of the fungicides used, ranging from 0.613 to 2.851 μg/mL and 0.104 to 4.073 μg/mL, respectively (Figure 1). EC50 for tebuconazole ranged from 0.413 to 7.779 μg/mL and sensitivity to prothioconazole varied from 0.658 to 10.396 μg/mL (Figure 1). The highest EC50 values were observed against azoxystrobin, which ranged from 0.293 to 12.495 μg/mL (Figure 1). No EC50 was calculated for *B. amyloliquefaciens* strain QST 713, as *S. sclerotiorum* isolates showed no growth at all concentrations. Within *S. sclerotiorum* isolates variability of EC50 values toward boscalid and fludioxonil was generally low and showed a less than four-fold difference in sensitivity between the greatest and the least sensitive isolate. In contrast, variability of EC50 values toward prothioconazole and azoxystrobin was high, where more than 10 to 12-fold differences in sensitivities were observed between isolates (Figure 1).

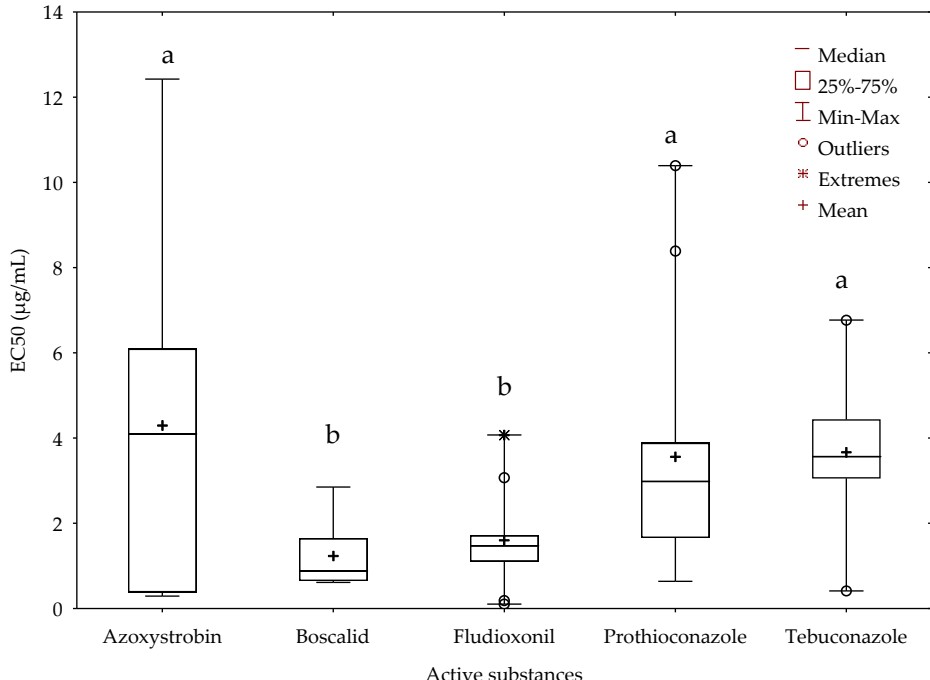

**Figure 1.** Distribution of EC50 values of *Sclerotinia sclerotiorum* isolates (*n* = 68) to various active substances. Data are presented as box-whisker plots with 25% and 75% quartiles (boxes), minimum and maximum values (whiskers), median (central line) and mean values of all isolates (plus sign). All circles above or below of each box plot represent outliers, stars represent extreme values. Means followed by the same letter were not different according to Fisher's least significant difference at $p \leq 0.05$.

3.2.2. Inhibition of Sclerotia Formation

With sclerotium formation on non-fungicide-amended PDA as the untreated control, all fungicides had inhibitory effects on the production of sclerotia and the weight of them (Figure 2). No sclerotium was formed on media amended with biopesticide *B. amyloliquefaciens* strain QST 713 (Figure 2). In other fungicide-amended media, the development of sclerotia was first increased up to concentration of 1 μg a.s. mL$^{-1}$, probably due to the fungicide stress, then the number and weight of sclerotia decreased with increasing of fungicides' concentrations (Figure 2). Almost complete suppression of sclerotia formation was achieved by fludioxonil, boscalid and prothioconazole at fungicide concentration of 30 μg a.s. mL$^{-1}$. Among all fungicides tested, azoxystrobin has the least effect on the sclerotia formation, in which a mean of 12 sclerotia per plate was also observed at concentration of 100 μg a.s. mL$^{-1}$ (Figure 2).

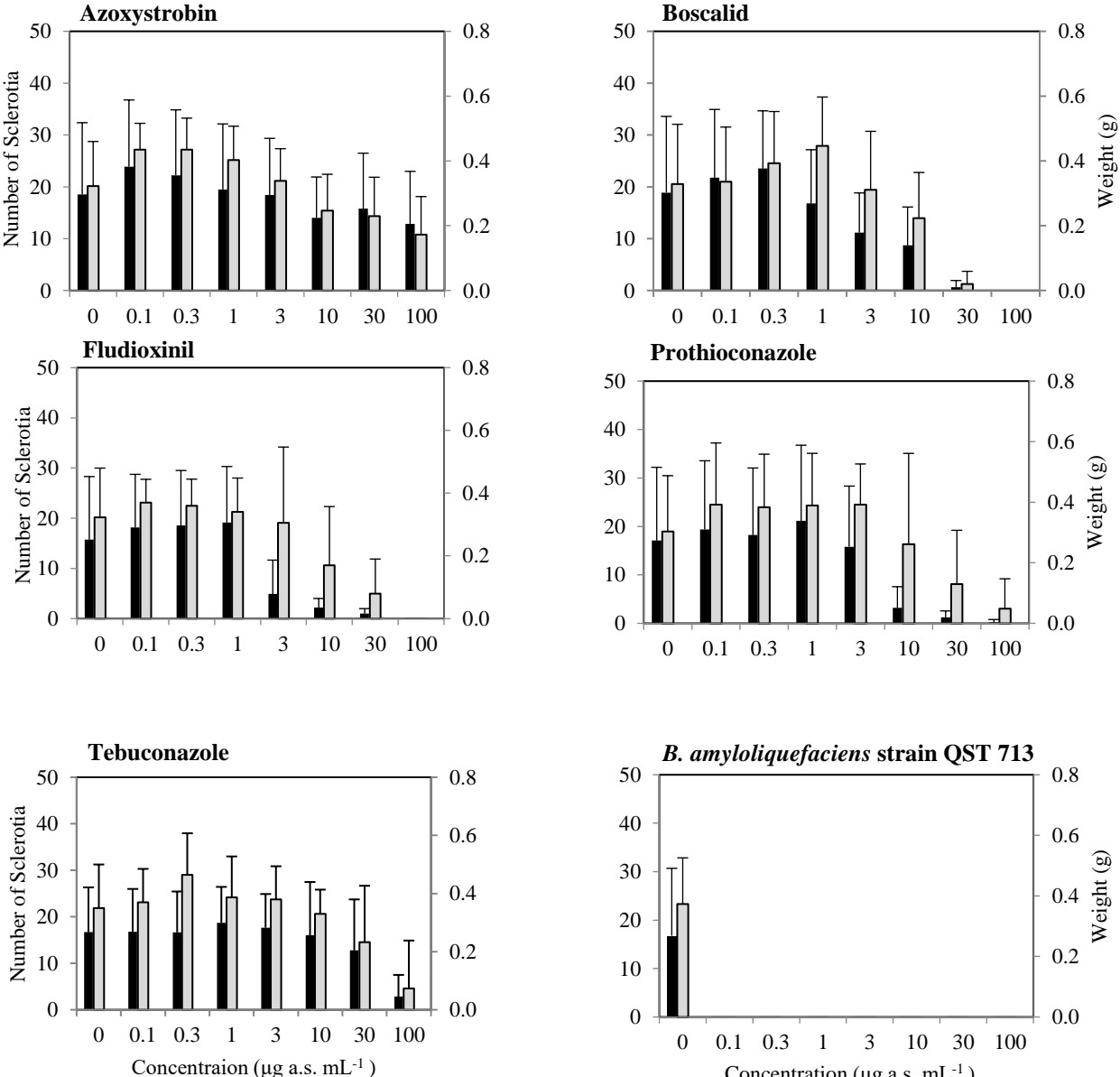

**Figure 2.** Effect of different fungicides on sclerotia formation (black bars) and the weight of them (grey bars). To evaluate the inhibitory effect of five fungicides as well as a biopesticide on production of sclerotia of *S. sclerotiorum* (*n* = 68) in-vitro, potato dextrose agar (PDA) was amended with the fungicides at seven concentrations. Plates incubated in 25 °C in darkness for 21 days for sclerotia formation. Each column indicate mean value ± standard deviations (error bars).

## 4. Discussion

To date, several studies indicated the presence of fungicide resistance or shift in sensitivity in *S. sclerotiorum* population toward different classes of active substances [10,14,23–25]. Consequently, on the one hand monitoring the emergence, incidence, and distribution of fungicide resistance or shift in sensitivity in *S. sclerotiorum* population should be carried out as a priority program, on the other hand, it is far essential to achieve competent fungicides with various modes of action to replace the compounds that are presently applied to control SSR. In addition, it is necessary to evaluate the combination of alternative modes of action to reduce selection for fungicide resistance to elevate disease management.

In the current study, the efficacy of a range of fungicides as well as a biopesticide, *B. amyloliquefaciens* strain QST 713 (former name: *Bacillus subtilis*), was evaluated against Sclerotinia stem rot in oilseed rape cultivation over two consecutive years. Significant differences were observed among the artificial inoculated and non-inoculated plots in both

growing seasons, indicating the benefit of applying inoculum to initiate disease. Additionally, it was shown clearly that the incidence of Sclerotinia stem rot is correlated with moisture periods before and during flowering [26]. Therefore, artificial inoculation with ground grain inoculum covered with Sclerotinia mycelia and mist-irrigation after inoculation led to strong development of SSR and severe infection up to 80% disease severity. Different compounds vary in their efficacy to control SSR under field conditions. Overall, all applications reduced 36.7% to 86.9% Sclerotinia disease severity, increased 55.2% to 98.7% yield, 1.5% to 7.0% thousand grain-weight (TGW), 1.5% to 5.9% oil content, and 0.1% to 1.3% protein content, respectively. However, application of fludioxonil (MoA: Inhibitor of MAP) and mixtures of boscalid/pyraclostrobin (SDHI + QoI) and fluopyram/prothioconazole (DMI + SDHI) were the most effective in reducing Sclerotinia disease severity, increasing yield and yield parameters. Combination of azoxystrobin/tebuconazole (DMI + QoI) exhibited a moderate level of control and azoxystrobin (QoI), tebuconazole (DMI) and boscalid (SDHI) were the least effective fungicides against SSR. What was apparent in the current study was that oilseed rape treated with the mixture of diverse modes of action led to considerable reductions of SSR and yield losses in comparison with solo compounds except fludioxonil. On the one hand, mixtures can be used to improve disease control; on the other hand, they can be used to manage resistance in pathogen population.

Related results were obtained by Kuang et al. [27] when evaluating the sensitivity of *S. sclerotiorum* to fludioxonil. They have seen that fludioxonil exhibits greater activity against Chinese *S. sclerotiorum* population than iprodione, which is a commonly applied fungicide in China. Furthermore, in agreement with the results of this study, Bradly et al. [28] reported that different fungicides such as azoxystrobin, benomyl, boscalid, iprodione, prothioconazole, tebuconazole, and vinclozolincan could be used effectively to control SSR and minimize yield reductions in oilseed rape in USA. However, it should be taken into consideration to rotate among the fungicide groups to slow down the selection for fungicide-resistant isolates. In their study, they have also observed that although fungicides are effective in reducing SSR incidence, yield reductions still do occur often, especially under heavy disease pressure. Similarly, Mueller et al. [23] demonstrated that when SSR incidence was high in soybean cultivations, no consistent control of disease was observed with benomyl or thiophanate methyl using different application systems. However, under low disease incidence, spray systems that were able to penetrate the canopy of soybean plants reduced the incidence of SSR down to 50% [23]. Contrary to the results obtained here, oilseed rape field experiments in China showed that the control efficacy of boscalid at 225, 300, and 375 g a.s./ha on SSR was 71%, 81%, and 90%, respectively, which was significantly higher than the control efficacy of carbendazim (belong to MBC fungicides) at 1500 g a.s./ha [10].

The results of present study also illustrate that the application of biopesticide *B. amyloliquefaciens* strain QST 713 could significantly suppress disease development, raise yield and yield parameters in comparison with inoculated untreated-controls. Nevertheless, the efficacy of *B. amyloliquefaciens* strain QST 713 was significantly lower than chemical treatments. Similar to this study, Hu et al. [29] indicated that application of *B. subtilis* Tu-100 not only significantly decreased the incidence of SSR in oilseed rape field trials artificially inoculated with *S. sclerotiorum* but also significantly increased plant vigor and seed yield. Yang et al. [26] explained that the efficacies of *B. subtilis* strain NJ-18 decreased slightly between the years due to the severe of stem rot in the field. Further study under field conditions demonstrated that the best biocontrol efficacy was obtained by spraying *B. subtilis* strain EDR4 at the same time as inoculation [30].

In the current study, the relative sensitivity of mycelial growth of *S. sclerotiorum* isolates to tested active substances demonstrated under in vitro conditions, did not have consistent predictive value with respect to performance of these fungicides under field conditions to control SSR. Similar results were observed in other studies, indicating that the in vitro studies only evaluated the interaction of the pathogen with each pesticide, whereas fungicide efficacy studies under field conditions incorporated the interaction of

the pesticide, fungal pathogen, and host plants, including the effect of the environmental conditions in which the host is living and where disease development occurs [31,32]. However, in evaluation of occurrence of fungicide resistance in pathogen populations toward pesticides, baseline sensitivity data are essential for assessment of emergence, occurrence and development of insensitivity within field populations of pathogens.

Of the five fungicides and a biopesticide evaluated under in vitro conditions, *B. amyloliquefaciens* strain QST 713, fludioxonil, boscalid considerably inhibit mycelial growth, and formation of sclerotia in *S. sclerotiorum* isolates. Fungal isolates were most sensitive to *B. amyloliquefaciens* strain QST 713, no mycelial growth and formation of sclerotia were observed over all amended petri plates. However, *B. amyloliquefaciens* strain QST 713 showed a low level of activity against SSR in field trials when compared to the other fungicides. Previous studies have shown that strains of *Bacillus* spp. grown on PDA produced antibiotic compounds that were released into the medium and inhibited hyphal growth of fungal pathogens [33–36]. Nevertheless, it should be taken into consideration that the successful development of microbial phase of a biological agent and production of antibiotics under natural field conditions relies upon numerous climatic factors and host plants, which increase or decrease the efficacy of biopesticide in comparison with the chemical fungicides.

Among all fungicides tested in the in vitro tests, the lowest EC50 values was observed towards boscalid in *S. sclerotiorum* population, however, boscalid showed a relative low level of efficacy against SSR in field trials. In agreement with our results, Ma et al. [24] reported that under in vitro conditions, boscalid showed greater inhibitory effect against *S. sclerotiorum* than dicarboximide fungicides. This proposes a differential synergic effect with the medium used or a comparable effect in the field experiments. The fungicides concentrations in planta in in vivo trials were unknown but may have been exceeded the concentrations used in the in vitro experiments.

Based on the EC50 values for 68 isolates used in this study, all *S. sclerotiorum* isolates showed higher variability in sensitivity toward azoxystrobin, prothioconazole, and tebuconazole (EC50 values range between 0.293 and 12.428 µg a.s. mL$^{-1}$) than boscalid and fludioxonil (with EC50 values of 0.104 to 4.073 µg a.s. mL$^{-1}$). Significant outliers in sensitivity toward prothioconazole and tebuconazole were also detected. One isolate exhibited extremely reduced sensitivity toward fludioxonil in comparison with other isolates of *S. sclerotiorum*. For discussed fungicides, except boscalid, in vitro and in vivo field trials yielded comparable results.

## 5. Conclusions

In conclusion, *Sclerotinia sclerotiorum*, the causal agent of Sclerotinia stem rot, is a devastating oilseed rape pathogen in Germany and fungicide application is one of the most promising methods to control the disease. The current study revealed significant differences in efficacy of different fungicides in regards to control of SSR as well as production of sclerotia as primary inoculum. Furthermore, to postpone the emergence and incidence of fungicide resistance, fungicides ought to be carried out in rotation and/or mixed with fungicides from classes having other modes of action or with low-risk fungicides to reduce the development of fungicide resistance in the *S. sclerotiorum* population.

**Funding:** This work was funded by the Julius Kühn-Institute and it did not receive any external funding.

**Data Availability Statement:** Data presented in this study are available upon request.

**Acknowledgments:** The author wants to thank the invaluable help of Jaroslaw Acalski, Kelly Coutinho Szinovatz and Anke Kawlath for their technical assistance.

**Conflicts of Interest:** The author declares no conflict of interest. The funder had no role in the design of the study; in the collection, analyses or interpretation of data; or in the writing of the manuscript.

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
