# Peer review of "Baseline Sensitivity and Control Efficacy of Various Group of Fungicides against Sclerotinia sclerotiorum in Oilseed Rape Cultivation"

_agronomy, doi:10.3390/agronomy11091758_

Round 1

Reviewer 1 Report

The paper of Dr. Zamani-Noor entitled “Baseline sensitivity and control efficacy of various group of fungicides against Sclerotinia sclerotiorum in oilseed rape cultivation”, has the aim of analysing the fungicide sensitivity of the Sclerotinia stem rot pathogen towards different fungicides commonly used in oilseed rape disease control strategies. The issue is essential to explore future challenges arising from the reduction of pesticide inputs in agriculture and the inclusion of antagonistic organisms in disease control strategies.

The paper is well written, English is fluent, methods are appropriate and results relevant, only some minor revisions are required.

Lines 70-76. In this paragraph on the introduction of biological agents, a few citations and explanations are needed to explain how different biological organisms compete or inhibit S. sclerotiorum and protect the crop from stem rot (e.g. producing antibiotics or enzymes, mycoparasitism, induced resistance).

Lines 99-100. Please specify which herbicides and insecticides were applied in the trial.

Line 103. How is a single isolate of S. sclerotiorum identified and confirmed as a unique isolate? How many isolates were from one field?

Lines 103-104. From which crops were the S. sclerotiorum isolates collected? Were these fields treated with pesticides? Please specify it in the manuscript.

Line 125 and Table 1. Why was Serenade ASO applied only once at application rate 2 L/ha? The full application rate is 8 L/ha for 6 times in the crop field. The low application rate could explain the higher disease severity in the plots.

Table 1. The producing company name should be added to the table and I suggest changing the "trade name" to "commercial product".

Line 196. Were the isolates collected from winter oilseed rape fields?

Line 197. As I understood the commercial fungicide products were used in fungicide sensitivity tests on PDA plates. Why these products were chosen and not pure active ingredients? 

Line 219. In which conditions were the sclerotia dried?

Table 2. In column DSI±SD, check the superscripts. In column TGW±SD, check the superscripts (b,c,bc) of the values 5.5±0.1 and 5.5±0.2.

Line 266 and Table 3. p-value should be presented <0.001, because p is not equal to 0.000.

Line 304. There is a duplication of prothioconazole and tebuconazole is missing from the list.

Table 4. Add ...inhibition±SD in the heading.

Figure 1. Please write "Tebuconazole" correctly on the figure.

Lines 399-409. An explanation of choosing the low application dose rate of Serenade ASO should be added. I suppose when the recommended dose of 8L/ha has been applied up to 6 times, the results were different. 

Reviewer 2 Report

This research reported the baseline sensitivity and control efficacy of various group of fungicides against Sclerotinia sclerotiorum in oilseed rape cultivation, which is very useful and applicable.  The manusript is well organized and written. However, some problems need to be addressed, please see attached comments on the manuscript.
